

**Effects of climate and forest composition on soil carbon cycling, soil organic matter stability and stocks in a humid boreal region**

**David Paré[1], Jérôme Laganière[1], Guy R. Larocque[1], Robert Boutin[1]**

[1]Natural Resources Canada, Canadian Forest Service, Laurentian Forestry Centre, 1055 du P.E.P.S., P.O. Box 10380, Stn. Ste-Foy, Quebec, QC G1V 4C7, Canada

**\*Correspondance:** David Paré  David.Pare@canada.ca

Research Topic: Vegetation Effects on Soil Organic Matter in Forested Ecosystems

**Keywords: soil respiration, litterfall, Q10, RS10, balsam fir, black spruce, soil organic matter turnover, long-term incubation**

10      ABSTRACT

The maintenance of the large soil organic carbon (SOC) stocks of the boreal forest under climate change is matter of concern. In this study, major soil carbon pools and fluxes were assessed in twenty-two closed-canopy forests located along an elevation and latitudinal climatic gradient expanding 4°C in mean annual temperature (MAT) for two important boreal conifer forest stand types:  balsam fir (*Abies balsamea*), a fire avoider and black spruce (*Picea mariana*), a fire tolerant species. SOC stocks were not influenced by climate or forest type. However, carbon fluxes, including aboveground litterfall rates as well as total soil respiration ($R_s$), heterotrophic ($R_h$), and autotrophic soil respiration ($R_a$) were linearly related to climate (cumulative degree days >5°C). The sensitivity of SOM

20      degradation to temperature, assessed by comparing $Q_{10}$ (rate of change for a T increase of 10°C) of soil respiration and $Rs_{10}$ (soil respiration rates corrected to 10°C) did not vary across the climate gradient, while the proportion of labile carbon and nitrogen showed higher values for balsam fir and for warmer sites. Balsam fir forests showed a greater litterfall rate, a better litter quality (lower C:N ratio) as well as a higher $Rs_{10}$ than black spruce ones, suggesting that their soils cycle a larger amount of C and N under a similar climate regime. Altogether, these results suggest that a warmer climate and balsam fir forest composition induce a more rapid SOC turnover. Contrary to common soil organic matter stabilization hypotheses, greater SOC cycling rates did not lead to higher total SOC stocks nor to the depletion of labile soil C and N. Positive effects of warming both on



fluxes to and from the soil as well as a potential saturation of stabilised SOC could explain
these results which apply to the context of this study: a cold and wet environment and a
stable vegetation composition along the climate gradient.

1.  INTRODUCTION

Of all the IPCC (Intergovernmental Panel on Climate Change) climate zones, the humid
boreal region contains the largest terrestrial carbon (C) stocks (380Pg). It is also the region
where the proportion of carbon contained in the soil is the greatest (Sharlemann et al.
2014). Boreal regions should also experience the most intense warming (Hoegh-Guldberg
et al. 2018). The consequences for the global climate of losing only a portion of the boreal
SOM stocks are potentially disastrous as this could create a highly damaging positive

feedback loop in which increasing warming would lead to greater soil C losses (DeLuca &
Boisvenue 2012). Recent assessments of global simulation models predicting the fate of
soil C in northern regions with global change showed that the model predictions vary
greatly between models and that they are prone to large uncertainties (Hunzinger et al.
2020; Wieders et al. 2019). These studies highlighted a great need to better understand the
SOM dynamics and its relation to climate and the necessity of empirical observations to
guide model development.

The use of climatic gradients to understand the impact of climate change on soil C
dynamics complements the information generated by soil warming experiments (Crowther
et al. 2016; Rustad et al. 2001). If vegetation types and soil conditions are held constant,

climatic gradients allow the observation of changes in soil fluxes with climate while
avoiding short-term pulse effects that have been observed in manipulative warming
experiments which may only be transient (Melillo et al. 2002). They can also be used to
observe changes that takes long time to form, including the buildup of stable SOM
(Lavallée et al. 2020).

Climate gradient studies generally show shorter organic matter turnover times under
warmer climate (Tewksbury and Van Miegroet 2007; Ziegler et al. 2017). However,
because both C fluxes to and from the soil are accelerated by an increase in temperature,
the net effect on soil C accumulation varies when the effect on one flux is greater than on
the other. For example, a Finnish thermocline study showed greater accumulation of soil C



in warmer climate (Liski and Westman 1997) due to increased productivity, while the
opposite was found for the SOM content of the soil organic layers of black spruce forests
of Alaska (Kane et al. 2005) and for both organic and mineral soil layers in continental US
(Garten 2012). Other studies revealed stable SOC stocks along gradients of increased mean
annual temperature in the Appalachian (Tewksbury and Van Miegroet 2007) and in
Newfoundland and Labrador (Ziegler et al. 2017). In the latter study, however, increased
fluxes to the soil at warmer sites were associated with higher C stocks in the organic layer.
Several studies that considered the decomposability of SOM have showed a higher content
of labile or poorly stabilized soil C in colder soils (Fissore et al. 2009; Ichikuza et al. 2007;
Laganière et al. 2015; Norris et al. 2011) suggesting that cold temperature maintains

reservoirs of easily decomposable SOM and consequently that cold soils may be more
susceptible to C losses upon warming.

The interpretation of climate gradient studies is often complicated by changes in vegetation
composition and in soil properties. The objective of the present study was to assess
changes in SOC stocks, quality and C fluxes to and from the soil along a climatic gradient,
while maintaining vegetation composition and soil properties as constant as possible for
two important boreal forest types: (1) forests dominated by balsam fir (*Abies balsamea* (L.)
Mill.), a fire-avoider that becomes abundant in landscapes with a long fire return interval,
and (2) forest dominated by black spruce forests (*Picea mariana* (Mill.) B.S.P.), a fire
adapted species that nevertheless can persist in the landscape in the absence of fire

(Couillard et al. 2018). Wildland fires are projected to increase in number, size and
intensity, due to climate-warming and fire frequency could increase by 1.5 to 4 times
before 2100 in the Canadian boreal forest (Boulanger et al. 2014), a situation that would
likely lead to a reduction in the area covered by balsam fir forests.

We hypothesized that:

- i) Stocks: Balsam fir sites, presumably because they produce greater amounts of
  litter of higher quality, as well as sites at the warmest end of the gradient should
  accumulate more stable SOM in line with the Microbial Efficiency-Matrix
  Stabilization (MEMS) framework developed in Cotrufo et al. (2013), which
  suggests that a larger inputs of labile plant material leads to larger and more stable





SOM stocks. A larger stock of stable SOM should be apparent in total C stocks as it represents a large share of total SOM (Andrieux et al. 2020).

- ii) Cycling: Warmer sites should exhibit enhanced C cycling to and from the soil.
- iii) SOM lability: Warm sites should contain less labile organic matter because low soil temperature protects labile SOM (Schmidt et al. 2011). The proportion of labile SOM can be evaluated with the use of long-term incubation. A lower proportion of labile SOM should also be reflected in a higher $Q_{10}$ for soil respiration rate according to the "C quality-temperature" (CQT) hypothesis (Laganière et al. 2015). Hence, balsam fir sites and warm sites should have greater soil C stocks and higher $Q_{10}$.

In comparison to other transect studies dealing with SOC dynamics, this study considered two forest types. Forest composition can have major impacts on SOC storage and cycling (Mayer et al. 2020). In addition, studying the impact of climate on SOC within vegetation type is important in the light of studies pointing out that vegetation shifts with climate change were found to be much more modest than expected. This has been related to non climatic factors including seed predation, pathogens and soil related limitations (Brown and Vellend 2014). In fact, land-use change and disturbances were found to have much greater effects on forest composition than climate change (Danneyrolles et al. 2019). These studies suggest that forest composition show a strong inertia to climate change and hence

that it is timely to study the impact of climate change within a vegetation type.



## 2. MATERIAL AND METHODS

Twelve balsam fir and ten black spruce sites were selected along a climatic gradient caused both by latitude and by elevation (Table 1; Fig. 1). All sites were mature closed-canopy stands with no signs of recent human or natural disturbances. The climatic gradient extended over a 4°C mean annual temperature difference, which represents MAT temperature increase expected for this region (2011-2080) according to IPCC scenario RCP4.5 (Côté et al. 2014). Sites were at least a few km apart from each other and are the

same sites as those described in Larocque et al. (2014) with the addition of three sites North of Lac St-Jean that were added to increase the number of cold black spruce sites. These sites were Lac Tirasse (TIR), RESEF (RES) and Eastern old black spruce (EOBS), a flux tower site that was intensively studied during the Fluxnet Canada and the Canadian Carbon program from 2002 to 2011. Only the closed-canopy section of this latter site (plots FCM 17-24) was considered and not the open lichen woodland portions (FCM 36-39) which would have make it too different from other black spruce sites along the climatic sequence. Four black spruce and three balsam fir sites were intensively studied. These sites are marked in bold in Table 1 and 2 (FM, FR, LS, J2, TIR, PR, OEBS). On these sites, root trenching was conducted to distinguish the contribution of autotrophic from that of

heterotrophic soil respiration and soil samples were collected for long-term laboratory incubations to determine the proportion of bioreactive (labile) soil C. Methods are described below. At each site, five 2 m x 2m sample plots where located on the perimeter of a 400 m$^2$ circular plot where forest tree mensuration measurements were conducted (see Larocque et al. 2014). In addition, on the seven intensive sites, five 2 m x 1 m root trenching plots (Tr) were installed, paired with each control plots. Field fluxes were measured during the snow free period for five years 2002-2006 at monthly interval during the snow free period, from early May to late October.



Table 1. Site characteristics

| Site | Lat. N | Long. W | Climate | Age (yr) | B.A. (m² ha⁻¹) | Soil type | Sa, Si and Cl (%) | MAT (°C) | Degree-days >5°C | Annual precip. (mm) |
|---|---|---|---|---|---|---|---|---|---|---|
| **Balsam fir sites** | | | | | | | | | | |
| **FM** | 47°19'00" | 71°06'00" | Cold | 70 | 44.6 | HF-P | 69, 20, 11SL | -1.0 | 856 | 1631 |
| RIVM | 47°18'18" | 71°08'00" | Cold | 60 | 49.6 | HF-P | 62, 29, 8SL | -0.1 | 953 | 1579 |
| RIVN1 | 47°21'09" | 71°06'07" | Cold | 50 | 45.4 | HF-P | 72, 22, 6SL | -0.5 | 903 | 1589 |
| RIVN2 | 47°19'42" | 71°06'01" | Cold | 50 | 56.5 | HF-P | 72, 22, 6SL | -0.1 | 954 | 1566 |
| SAUT | 47°19'06" | 71°12'46" | Cold | 50 | 46.1 | HF-P | 69, 22, 9SL | -0.7 | 874 | 1634 |
| **FR** | 47°08'17" | 71°17'08" | Mild | 75 | 31.2 | H-P | 75, 21, 4SL | 0.9 | 1109 | 1518 |
| KM82 | 47°09'36" | 71°15'20" | Mild | 53 | 45.9 | HF-P | 75, 17, 8SL | 0.9 | 1101 | 1514 |
| PTR | 47°07'00" | 71°20'00" | Mild | 42 | 44.7 | FH-P | 66, 28, 6SL | 2.0 | 1287 | 1448 |
| **LS** | 46°52'42" | 71°43'17" | Warm | 50 | 47.7 | HF-P | 70, 20, 10SL | 3.3 | 1536 | 1344 |
| STMAT | 47°03'24" | 71°39'41" | Warm | 50 | 38.9 | HF-P | 79, 18, 3LFS | 3.3 | 1523 | 1407 |
| SLAC7 | 46°55'34" | 71°46'51" | Warm | 45 | 33.6 | HF-P | 39, 52, 9SiL | 2.5 | 1395 | 1422 |
| DUC-2 | 46°56'05" | 71°39'15" | Warm | 62 | 35.1 | HF-P | 47, 39, 14L | 3.3 | 1536 | 1367 |
| **Black spruce sites** | | | | | | | | | | |
| J1 | 47°28'05" | 71°14'10" | Cold | 70 | 36.3 | FH-P | 43, 52, 5SiL | -0.6 | 904 | 1588 |
| **J2** | 47°28'08" | 71°13'47" | Cold | 65 | 45.7 | FH-P | 74, 16, 10SL | -0.6 | 904 | 1588 |
| J3 | 47°28'09" | 71°13'47" | Cold | 70 | 37.6 | FH-P | 66, 34, 0SL | -0.6 | 904 | 1588 |
| CJC | 47°36' | 71°14'00" | Cold | 103 | 45.5 | H-P | 67, 29, 4SL | -0.5 | 926 | 1540 |
| **TIR** | 49°12'27" | 73°38'03" | Mild | 84 | 32.7 | HF-P | 84, 13, 3 S | -0.1 | 1139 | 1042 |
| **EOBS** | 49°41'33" | 74°20'31" | Mild | 120 | 17.2 | FH-P | 68, 28, 4SL | -0.3 | 1195 | 961 |
| RES | 48°48'41" | 72°46'03" | Mild | 82 | 34.2 | HF-P | 76, 20, 4LS | 0.8 | 1298 | 954 |
| AVRP | 46°56'20" | 72°07'40" | Warm | 48 | 32.1 | HF-P | 77, 15, 8SL | 3.0 | 1524 | 1229 |
| REPO | 47°01'34" | 72°07'02" | Warm | 46 | 44.3 | FH-P | 70, 26, 4SL | 3.0 | 1525 | 1199 |
| **PR** | 46°48'48" | 71°36'20" | Warm | 50 | 23.2 | HF-P | 62, 29, 9SL | 3.6 | 1601 | 1293 |

Note: Temperature and precipitation values are averages over a 30-year period computed using BioSIM (Régnière, J. and R. Saint-Amant, 2008). B.A.:basal area; Sa, Si, Cl: sand, silt and Clay respectively; Humo-Ferric Podzol (HF-P); Ferro-Humic Podzol (FH-P) or Humic Podzol (H-P); Textural class: Sandy Loam (SL), Loamy fine Sand (LFS), Silty Loam (SiL), Loam (L) Podzol (H-P).



Fig. 1. Location of the study sites showing how the climate gradient is related to elevation and latitude. Data from Abrams et al. (2020).

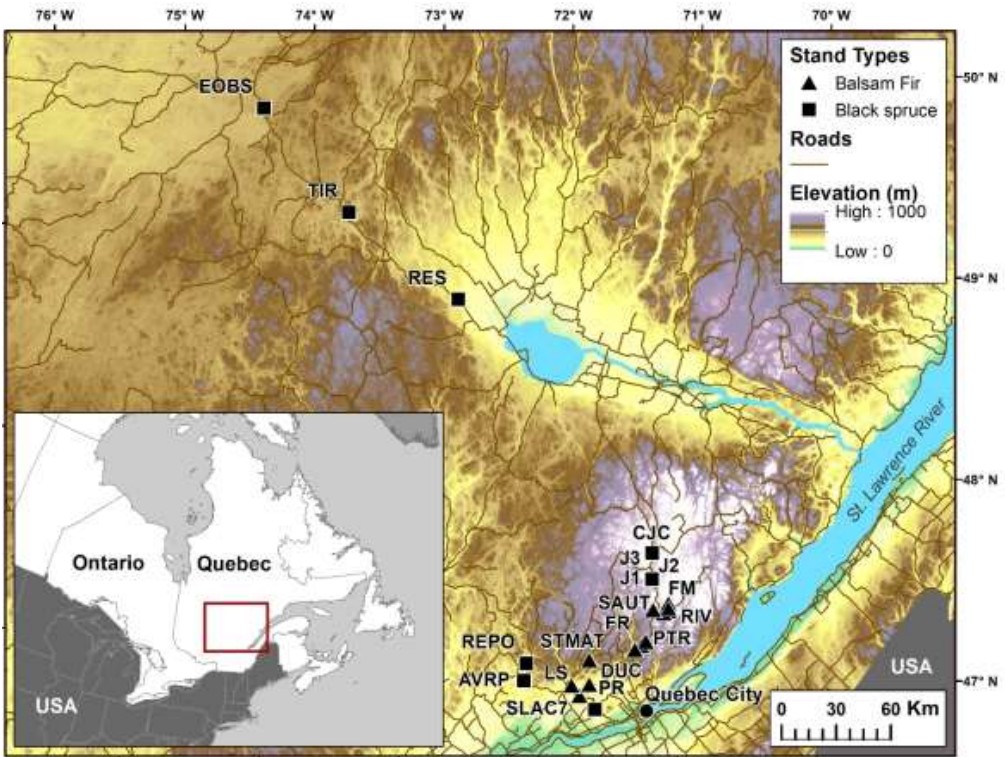

## 2.1 Soil C pools

At each sample plot, five volumetric soil samples of the organic layer (FH layers) were sampled using a 400 cm² template and five samples of the top 0-20 cm mineral soil layer were sampled with a 4.70 cm diameter corer. This generated 25 soil samples per layer and site. In addition, one 20-40 cm volumetric mineral soil sample was taken per sample plot. The exact depth of the sampling was noted in the cases when it was not possible to sample to these depth because of rocks. Samples where kept in a cold room at 3°C until being processed. Then they were oven dried at 70°C and sieved at 2mm to remove roots and coarse fragments. They were then weighted and C content was determined by dry combustion using a LECO CNS-2000 analyzer (Leco Corp., St. Joseph, MI, USA).



## 2.2 Litterfall

Eight square littertraps (2862 cm$^2$) were installed on each site. Litterfall was collected at least twice a year during three years. Litterfall per trap was divided by the number of days for the full sampling period and then multiplied by 365 to get annual estimates. Samples were kept at 3ºC,before being sorted to distinguish foliar material from other fine material (small branches, mosses, insects, insect frass, cones, flowers and unknown materials) dried at 70ºC. They were ground on a Wiley-mill and analysed for C and nitrogen (N) content as above.

## 2.3 Soil respiration *in situ*

In each sample plot, both in control and trenched plots, two 5 cm deep polyvinyl chloride
(PVC) collars measuring 10 cm in diameter were inserted into the soil at a 1m distance from each other (10 collars per site) to measure total soil respiration ($R_S$). On the seven intensive sites, trenches were dug to a depth of approximately 30cm on a 2 x 1m area near each sample plot to sever roots. Trenches were lined with landscaping fabric and backfilled. Herbs and shrubs were removed from trenched plots periodically. Two PVC collars were installed in each plot to measure heterotrophic soil respiration ($R_h$). Installation was conducted in June and measurements started the following year to let the severed roots decompose and avoid a method-induced $CO_2$ peak. $CO_2$ efflux measurements were performed once a month from early May to mid-late-October. They were measured by placing the soil chamber (LI-6400-09) of a portable infrared gas analyzer (LI-6400, LI-COR Inc., Lincoln, NE) above the collar.
During soil $CO_2$ efflux measurements, soil temperature at a depth of 10 to 15 cm was recorded as well as soil moisture (i.e. volumetric water content, % v/v) using a TDR-300 time domain reflectometry moisture probe (Spectrum Technologies Inc., Plainfield, IL). This relatively shallow depth was chosen because soil temperature at this depth fluctuates daily along the growing season. Soil temperature was measured continuously on the intensive sites with thermocouple wires attached to a Campbell Scientific data station, while WatchDog 100 Series Water Resistant Button Loggers were installed on the other sites. As detailed in Table 5, 5093 $CO_2$ efflux soil collar measurements were performed on control plots and 2661 on trenched plot over the duration of the study. Measurements were performed over 4 years for most sites; three sites were sampled for only 3 years, and one site for two years at a rate



of five to six measurement days per year (details in Table 5). Several parameters were derived from the respiration vs temperature relationship. The relationship of soil CO2 efflux vs temperature was parametrized with a simple exponential equation (Buchmann 2000):

[Eq.1]: $Rs = b_1 \, e^{(b2 \cdot (Ts-10)}$

Where:

Rs is soil respiration in $\mu$moles C m$^{-2}$ $*$ s$^{-1}$;

b1 and b2 are coefficients and Ts is soil temperature in $^{\circ}$C at a depth of 10-15cm.

$Q_{10}$, the rate of change:

[Eq.2]: $Q_{10} = e^{(10*b2)}$

$RS_{10}$, the soil respiration at 10$^{\circ}$C is

[Eq.3]: $Rs_{10}: b_1 * Q_{10}$

The parameters of the equations were estimated for each site using all measurements from all years. We combined all the measurements per site because we noted that the fit was better when the range in soil temperatures used was wider and because we noted the trends were

210 comparable between years.

2.4 Assessing annual soil respiration fluxes

We estimated soil respiration from May 1$^{st}$ to October 31$^{st}$ (183 d) period. At the first measurement period, soil temperature was close to 0$^{\circ}$C and at the last measurements; it varied from 2 to 4.5$^{\circ}$C depending on local climate. This period is close to the snow free period. Ecosystem respiration during the snow-covered period has been estimated at about 10% of total ecosystem respiration for one of our site (Bergeron et al. 2007). The methodologies outlined in Lavigne et al. (2003) were used to scale CO$_2$ efflux over the year (183 d) and to derive CO$_2$ effluxes between measurement dates and for a few days at the end

or beginning of the season. This method accounts for seasonal trends in Rs that are not related





to temperature change. For example, changes that may be due to a seasonal flush of labile C
with fine root development or with fresh litter fall inputs. Briefly, a $RS_{10}$ value was calculated
for each site and measurement date assuming a $Q_{10}$ value of 2 as discussed in Lavigne et al.
(2003). This value was chosen because most model use this value (Meyer et al. 2018) and
because we have no knowledge if the $Q_{10}$ value that was estimated for the whole growing
season is appropriate for specific periods within the season. A $Rs_{10}$ value was then linearly
interpolated for each calendar day between each measurement day. This daily $Rs_{10}$ value was
used, together with the recorded soil temperature at 10-15cm depth, to estimate daily soil
respiration per site with equation 4. This equation converts the units from $\mu mol\ CO2.s^{-1}.m^{-2}$
into kg C per ha per day assuming a $Q_{10}$ of 2. Daily measurements were summed from May
1st to October 31th to obtain yearly estimates of Rs.

[Eq.4]:  $Rs = Rs_{10}*2.7183^{(0.0693*(Ts-10))}*3600*24*12*10^7$

2.5 Partitioning into heterotrophic and autotrophic respiration
On the intensive sites, the same calculation as above was performed for the trenched plots
to obtain an annual soil efflux per site for the trenched plot. The same parameters were
fitted using Eq 1 to 3. The parameters derived from these equations were analysed and
discussed. However, because the values were higher than those generally reported in the
literature (Rh averaged 77% of Rs) although close to the values reported by Lavigne et al.
(2003) for cold sites, we suspected that the trenching was incomplete. We therefore used
the often-used (Naidu & Bagchi 2021) relationship of Bond-Lamberty et al. (2004):

[Eq.5]:  $ln\ (Rh)= 1.22+0.73\ ln(Rs)$

Where R (h or s) is respiration in $g\ C\ m^{-2}\ year^{-1}$. This value was subtracted from Rs to
estimate Ra, autotrophic respiration.

2.6 Lab incubation
For the microcosm experiment, soil samples from four of the five plots per site were
pooled by depth to yield composite samples for four intensively studied sites (Mild-Fir:





FR; Warm-fir: LS; Cold-Spruce: J2 ; Warm-Spruce: PR; see Table 1 for site codes).
Organic and mineral layer samples, 6- and 40-g dry weight, respectively, were placed on a
layer of glass wool in 120 mL plastic containers (28 cm$^2$ surface area), wetted at field
capacity and placed in 500 mL glass jars (Mason type). These microcosms were left to
stabilize for one week at 2ºC after handling and were then incubated in the laboratory at 3,
10, 15 and 22°C at field capacity for 354 days. These temperatures were chosen to cover
the range of soil temperatures that decomposers would experience in the field. Soil
respiration rates were measured monthly over 4 to 24h intervals depending on rates and

CO2 was measured on LI-6400 portable photosynthesis system (LI-COR, Lincoln, NE,
USA) using the method described in Andrieux et al. (2020). At each measurement period,
samples were flushed with distilled water and NO3-N and NH4-N were analyzed by FIA
(Quickchem 8500, Lachat Instruments, Loveland, Colorado). Using linear interpolation
between measurement dates, the total amount of C and N mineralized during the full
incubation was estimated and considered as labile C or labile N.

### 2.7 Statistical analysis

The effects of climate (DD), species and their interaction on the dependant variables (soil
carbon pools and fluxes as well as soil respiration parameters ($Rs_{10}$, $Q_{10}$) and litterfall
properties) were tested using proc GLM (SAS 2013). DD being a continuous variable and

species a categorical one. Sites are considered as the experimental units. For laboratory
incubation, the effects of incubation temperature, climate, and species were tested using
linear regression models fitted separately on each  soil layer. Due to the small sample size,
interactions between independent variables could not be evaluated. Effects were considered
significant at the 0.05 threshold level.

## 3. RESULTS

### 3.1 Soil carbon stocks

Soil C stocks varied widely among sites, ranging from 108 to 202 C t.ha$^{-1}$ for fir sites and
from 76 to 310 t.ha$^{-1}$ for spruce sites (Table 2). This variability could not be attributed to a

single factor. The only site with particle size distribution classifying it as a sand, TIR, had a





low C stock, but other sites that were not as sandy, also showed low C content (Tables 1 & 2). On balsam fir sites, the humus layer contained an average of 25% of estimated soil C stocks, while this proportion was 37% on spruce sites. Excluding the two sites where the 20-40 cm layer could not be sampled, the latter value falls to 33% and is thus close to that found for the balsam fir sites. No significant effects of tree species, climate and their interaction on soil C stocks were found (Table 3; Fig. 2a). These results are for the organic layer (FH) and the top 0-20cm mineral soil layer.





Table 2. Soil carbon stock (t C ha$^{-1}$) mean and coefficient of variation (CV) for the study sites. O.L. stands for organic layer (FH) and 0-20 cm and 20-40cm for the soil depth increments of mineral soil samples. *CWD: Coarse woody debris.

| Site ID | Species | Climate regime | O.L. mean | CV | 0-20 cm mean | CV | 20-40 cm mean | CV | CWD* mean | CV | Total C mean | CV |
|---|---|---|---|---|---|---|---|---|---|---|---|---|
| **Balsam fir sites** | | | | | | | | | | | | |
| **FM** | BF | Cold | 27.6 | 0.19 | 40.7 | 0.26 | 33.8 | 0.16 | 5.5 | 0.28 | 107.6 | 0.21 |
| RIVM | BF | Cold | 30.6 | 0.30 | 64.6 | 0.27 | 44.9 | 0.26 | 4.4 | 0.85 | 144.5 | 0.29 |
| RIVN1 | BF | Cold | 23.5 | 0.20 | 63.5 | 0.33 | 57.7 | 0.18 | 10.0 | 0.78 | 154.6 | 0.28 |
| RIVN2 | BF | Cold | 44.8 | 0.45 | 55.9 | 0.35 | 39.0 | 0.58 | 12.3 | 1.31 | 152.1 | 0.52 |
| SAUT | BF | Cold | 40.3 | 0.35 | 73.1 | 0.31 | 43.4 | 0.14 | 7.6 | 0.50 | 164.4 | 0.28 |
| **FR** | BF | Mild | 53.3 | 0.27 | 84.9 | 0.23 | 52.4 | 0.26 | 11.2 | 0.64 | 201.7 | 0.27 |
| KM82 | BF | Mild | 40.7 | 0.38 | 71.5 | 0.31 | 47.9 | 0.13 | 9.1 | 0.71 | 169.2 | 0.30 |
| PTR | BF | Mild | 37.7 | 0.50 | 50.2 | 0.63 | 30.3 | 0.13 | 4.8 | 0.57 | 122.9 | 0.46 |
| **LS** | BF | Warm | 37.6 | 0.21 | 68.5 | 0.22 | 47.7 | 0.17 | 15.0 | 0.98 | 168.8 | 0.27 |
| STMAT | BF | Warm | 50.0 | 0.36 | 48.8 | 0.42 | 45.6 | 0.18 | 3.4 | 0.74 | 147.8 | 0.33 |
| SLAC7 | BF | Warm | 32.7 | 0.57 | 68.3 | 0.30 | 41.8 | 0.24 | 4.6 | 1.58 | 147.3 | 0.38 |
| DUC-2 | BF | Warm | 49.1 | 0.43 | 60.7 | 0.37 | 54.6 | 0.20 | 6.6 | 1.06 | 171.0 | 0.36 |
| **Black spruce sites** | | | | | | | | | | | | |
| J1 | BS | Cold | 33.0 | 0.32 | 42.9 | 1.05 | n.a | n.a | 0.1 | 1.19 | 76.1 | 0.73 |
| **J2** | BS | Cold | 51.2 | 0.18 | 84.0 | 0.38 | 41.9 | 0.61 | 3.9 | 0.87 | 181.0 | 0.39 |
| J3 | BS | Cold | 57.1 | 0.33 | 78.4 | 0.50 | 124.7 | 0.49 | 2.5 | 0.43 | 262.7 | 0.46 |
| CJC | BS | Cold | 100.5 | 0.26 | 66.6 | 0.59 | 96.6 | 0.96 | 11.9 | 0.78 | 275.7 | 0.61 |
| **TIR** | BS | Mild | 39.5 | 0.25 | 42.4 | 0.30 | n.a | n.a | n.a | n.a | 81.9 | 0.28 |
| **EOBS** | BS | Mild | 63.6 | 0.35 | 17.1 | 0.5 | 24.7 | 0.64 | n.a | n.a | 105.4 | 0.45 |
| RES | BS | Mild | 89,8 | 0,50 | 52,1 | 0,29 | n.a | n.a | n.a | n.a | 141,9 | 0,43 |
| AVRP | BS | Warm | 18.1 | 0.54 | 74.3 | 0.50 | 25.4 | 0.75 | 9.5 | 1.29 | 127.2 | 0.61 |
| REPO | BS | Warm | 121.3 | 0.34 | 60.4 | 0.61 | 119.0 | 0.75 | 9.3 | 0.50 | 310.0 | 0.56 |
| **PR** | BS | Warm | 30.5 | 0.19 | 65.7 | 0.14 | 21.7 | 0.26 | 12.1 | 0.49 | 130.1 | 0.21 |





Table 3. Results of statistical analyses (p values) from a GLM testing for forest type (tree species), degree-days (DD) and their interaction for C fluxes, soil C pools and estimated parameters from the *in situ* soil respiration - soil temperature relationships.

| Response variables | description | tree species | DD | interaction |
|---|---|---|---|---|
| FH-TotC | C mass of the FH layer (t C ha$^{-1}$) | 0.1061 | 0.7854 | 0.5984 |
| Soil-TotC | C mass of the soil down to 20cm (t C ha$^{-1}$) | 0.6495 | 0.9281 | 0.9770 |
| FolLit | Annual foliar litter (kg C ha$^{-1}$ yr$^{-1}$) | **<0.0001** | **0.0013** | 0.5830 |
| Lit | Annual litter (kg C ha$^{-1}$ yr$^{-1}$) | **0.0014** | **0.0066** | 0.7020 |
| Lit C :N | Foliar litter C:N (unitless) | **0.0003** | 0.3807 | 0.2906 |
| Rs (183days) | Soil respiration (kg C ha$^{-1}$ yr$^{-1}$) | 0.4390 | **0.0001** | 0.1383 |
| Rs$_{10}$ | Respiration at 10$^o$C  (µmol m$^{-2}$ s$^{-1}$) | **0.0472** | 0.9266 | 0.0590 |
| Q$_{10}$ | Change in respiration rates (unitless) | 0.2347 | 0.8892 | 0.8976 |





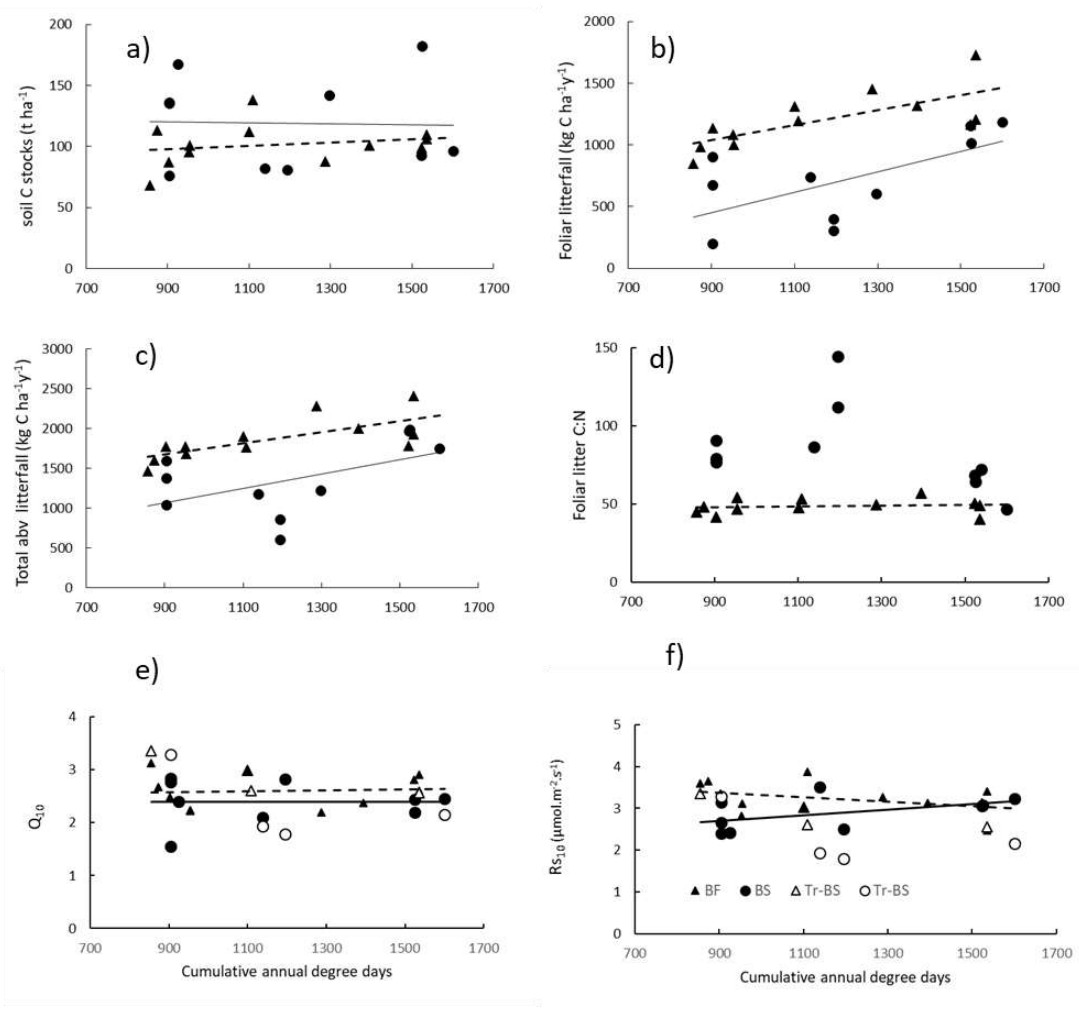

Fig.2. (a) Soil carbon stocks (humus + top 0-20cm mineral soil) (b), soil annual foliar litterfall, (c) total annual aboveground litterfall, (d) foliar litter C:N ratio, (e) $Q_{10}$ (e) and $RS_{10}$ (f) derived from field soil respiration measurements, as related to forest composition and cumulative annual degree-days (DD). Only the relationships with litterfall (b,c) were significant for DD effects (see table 3). Legend: Triangles are for fir stands and circles for spruce stands; BS stands for black spruce and BF for balsam fir; Tr stands for trenched plots and are identified by open symbols.



3.2 Carbon fluxes to and from the soil

Aboveground litterfall was linearly related to degree-days as well as to tree species, but the
interaction of these two factors was not significant (Table 3; Fig. 2b, c), indicating that the
slope of litterfall with degree-days did not differ between the two forest types. On average,
fir forests produced 71% more foliar litter and 37% more total litter than black spruce ones.
The N concentration of foliar litter was higher for balsam fir than for black spruce and this
is reflected in lower C:N values for the former (Table 3; Fig.2d). This variable was
remarkably stable for balsam fir across the climate gradient, while it showed greater
variability for black spruce but no trend with climate (Fig. 2d).

Cumulative soil respiration from May 1$^{st}$ to October 31$^{st}$ was linearly related to degree-
days and was not significantly affected by tree species (Table 3, Fig. 3, SM Table 1).
Autotrophic, heterotrophic and total soil respiration rates were strongly and linearly related
to degree-days (Fig.3). The slope of autotrophic respiration with degree-days was greater
than that of heterotrophic respiration (SM Table 1), although this difference was not
significant at the 5% threshold (results not presented; p=0.0832), suggesting that a higher
share of autotrophic respiration to total soil respiration for warm sites as observed by
Lavigne et al. (2003). For two of the coldest sites, the values obtained from trenched plots
(indicated by stars in Fig. 3) were similar to the value estimated with the Bond Lamberty et
al. (2004) equation. However, the field values obtained for the warm sites were much
higher than the modeled values. This could indicate that the trenching of the warm sites
was incomplete.





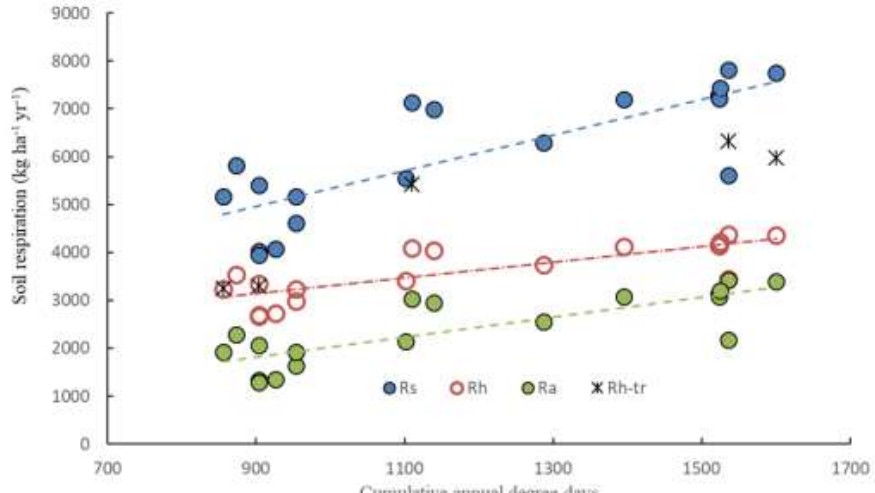

Fig.3. Seasonal soil respiration (183 days: May 1st- October 31st) with cumulative degree days for Rs (total soil respiration), Rh (heterotrophic soil respiration) and Ra (autotrophic soil respiration); stars indicates heterotrophic respiration assessed for trenched plots. Rh and Ra are otherwise calculated using the equation of Bond-Lamberty et al. 2004. All slopes are significant at the 0.001 level. More information in SM Table 1.

### 3.3 Reactivity of SOM to temperature and tree species

Soil respiration measured *in situ* was dependant on soil temperature but the fit of the $Q_{10}$ relationships varied greatly between sites. $R^2$ values ranged from 0.17 to 0.62 and averaged 0.43. The $Q_{10}$ values ranged from 1.55 to 3.12 and averaged 2.51. $Rs_{10}$ varied from 2.41 to 3.87 and averaged 3.09 $\mu mol.m^2.s^{-1}$ (SM Table 2). The effect of climate (degree-days) on both $Q_{10}$ and $Rs_{10}$ was not significant (Table 3; Fig 2e,f). This indicates that colder sites did

not show a greater reactivity of soil respiration to temperature and that at a similar temperature, cold sites would not show a greater soil respiration than warm ones. $Rs_{10}$ was significantly affected by tree species while the interaction with climate was not significant (Table 3). These results suggest that balsam fir soils would show a greater soil respiration rate than black spruce ones under a similar climate. On trenched plots, the values of $Rs_{10}$ ranged from 1.94 to 3.36 and averaged 2.53 and were, as expected, systematically lower as compared to their respective control plots (Table 3 Fig,2 f: empty symbols). The $Q_{10}$ values of trenched plots ranged from 1.55 to 3.12 and averaged 2.51 across sites and were not




systematically lower nor higher than the values of their respective control plots (Table 3
Fig,2 e).  Laboratory incubation indicated a higher proportion of labile C and N in the

humus than in the mineral soil (SM Table3). Incubation temperature had an overall
significant effect for both soil layers and elements. Climate and species had an effect only
for the organic layer (SM Table 3). Warm and balsam fir sites showed consistently higher
N and C mineralization rates (Fig. 4). This effect was significant at the 0.05 threshold for N
but barely missed that mark ($p=0.06$) for species effect on C mineralisation (SM Table 3).

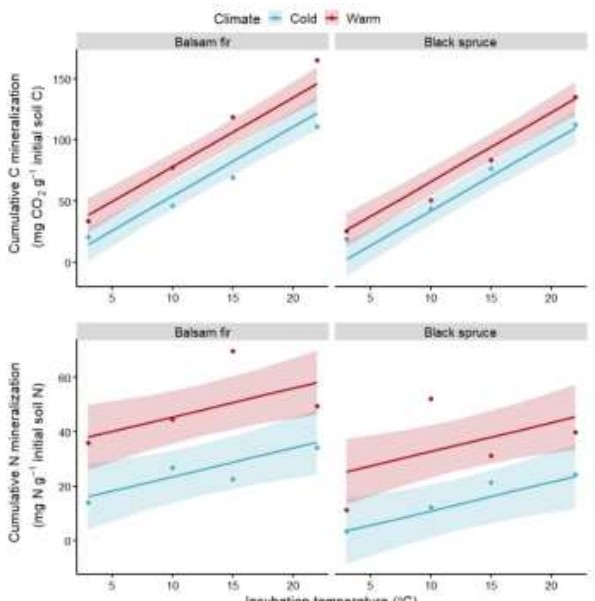

Fig. 4 Fitted relationships for cumulative soil $CO_2$-C and N mineralization with incubation

temperature, site climate (warm vs. cold) and tree species (balsam fir vs. black spruce) in
the organic layer.  Red colour identifies warm sites and blue colour cold sites, with points
showing observed values. Shaded areas indicate 95% confidence intervals.




## 4. DISCUSSION

### 4.1 Soil Carbon stocks

No significant changes in SOC stocks along the climate gradient were found and we refute hypothesis i. Our result do not give support to the MEMS framework (Cotrufo et al 2013). A warmer climate increases litterfall inputs as well as in soil respiration rates without

causing a net effect on C stocks. However, we must be cautious with this interpretation because soil C stocks estimates have a high level of uncertainty, and it is not possible, given our sampling effort to detect small changes (Yanai et al. 2003). Climate gradient studies in the boreal region have shown that warmer conditions can increase (Liski, & Westman, 1997), have no effect (Ziegler et al. 2017) or lower soil carbon stocks stocks (Norris et al. 2011). This latter study was conducted in jack pine forests of central Canada, a region that has a much drier climate than the one of this study and of that of the other studies cited above. When heat coincides with drought, which is more likely in drier climates, these events strongly reduce gross primary production (GPP) but yield smaller reduction in ecosystem respiration, leading to strong reduction in net ecosystem

productivity (NEP) (Von Buttlar et al. 2018). However, under the cold and wet conditions at our sites, microbial decomposition as well as plant productivity are strongly temperature-limited and greater litter inputs due to warmer conditions are likely offset by a greater microbial decomposition.

Our warmer sites generally had lower precipitation (Table 1). However, soil moisture measured during $CO_2$ efflux measurements did not show consistent difference across the gradient (data not shown). In addition, our study area does not suffer from water deficit and tree growth is not sensitive to summer soil moisture (Girardin et al. 2021). The size of the SOM stock is not only controlled by climate or net primary production (NPP) and is strongly influenced by soil types and drainage (Andrieux et al. 2018, Dalsgaard et al.

2016). While we tried to maintain these conditions as constant as possible in our site selection, they may still play an important control on soil C stocks overriding that of C fluxes (Dalsgaard et al. 2016). We were not able to explain the large variability in soil C stocks across sites. This property is highly variable at small scale and has notoriously been difficult to map (Paré et al. 2021). A much larger dataset would be required to test for the





effect of environmental factors on soil C storage and should include gradients of texture, drainage, vegetation composition and climate with as much as possible independence between driving variables.

4.2 Carbon fluxes in relation to forest type and climate

Carbon fluxes to and from the soil were all linearly related to the cumulated degree-days in agreement with hypothesis ii. Litterfall rates was the only parameter that also showed a clear species effect. This may be due to the fact that fir trees do not keep their needles for as long as black spruce; respectively 4 to 5 years compared to a maximum of 10 years. Both species also showed a stable litter C:N ratio along the climate gradient, suggesting that the stoichiometry of C to N is not affected by climate. The difference in litter C:N ratio between species is not surprising because black spruce typically show lower foliar N concentration than balsam fir (Paré et al. 2013). Despite this difference for N, and perhaps greater N limitation in black spruce forests, the C litterfall rate of both species reacted positively and with the same slope to a warmer climate. Litterfall has been found to be a

good predictor of net primary production (Mahi et al. 2011). Our results thus suggest an enhanced tree productivity under warmer conditions. This result contrasts with those of Larocque et al. (2014) conducted on approximatively the same sites, which did not show any significant differences in radial tree growth along the climate gradient. This lack of effect could be explained by the fact that the forest stands were mature, at a stage where growth is slow. D'Orangeville et al. (2016) using a much larger dataset did observe a positive effect of warming on tree growth in Quebec for both balsam fir and black spruce. They predicted that enhanced tree growth would be observed for an increase of 4°C even without a change in precipitation for black spruce but would require a 15% increase in precipitation for balsam fir.

The linear trends that we observed between carbon fluxes and climate using a relatively small number of sites indicate that the assessment of biogeochemical fluxes, including soil respiration measurements or foliar litterfall may be more suited to study changes in NPP than the often-used tree ring analysis. These fluxes are less affected by stand maturity and are more directly connected to carbon capture by photosynthesis than stemwood biomass, which only represents from 10 to 30% of photosynthate allocation and depends on many





physiological processes including allocation to below and aboveground parts, reproduction, defense and energy storage (Litton et al. 2007). Our results showed clear and linear trends with warming for litterfall and soil respiration, while tree ring analysis have shown divergent results in the Canadian boreal forest (e.g. Girardin et al. 2016 vs Hember et al.

2019).

Increased litterfall rate with a warmer climate was accompanied by a linear and positive response of soil respiration indicating a greater turnover rate of soil C as the climate warms. These results are coherent with the observations of Ziegler et al. (2017) for a climatic gradient in balsam fir forests in Newfoundland and Labrador and suggest that soil C stocks can be maintained despite a greater turnover rate. These results are in line with an acceleration of C turnover rate observed at the global scale over the recent decades (Carvalhais et al. 2014). Interestingly, congruent results, that is to say maintenance of soil OC pools despite a faster turnover, have also been observed for a tropical mountain climatic gradient (Giardina et al. 2014) suggesting that within a biome where water

limitation to forest production is of minor importance, the same pattern of accelerated C cycling and maintenance of pools may occur.

4.3 Labile SOM in relation to a warmer climate

Several studies have concluded that a warmer climate would lead to a depletion of labile or poorly stabilized SOC (Fissore et al. 2009; Ichikuza et al. 2007; Laganière et al. 2015; Norris et al. 2011). Our results indicated to the contrary, that the lability of SOC is either similar or higher with a warmer climate. Field results revealed that Q10 and Rs10 were stable along the climate gradient, while results from the laboratory incubation, indicated a greater proportion labile C and N for balsam fir and for warmer climate but only for the O

layer. Both results support the rejection of hypothesis iii. The stability of Rs10 along the climate gradient suggests that labile soil organic matter does not accumulate more on cold than on warm soils. The higher Rs10 values found for balsam fir soils, irrespectively of climate, are congruent with the greater litterfall rates observed for this species in this study.



According to the "C quality-temperature" (CQT) hypothesis, decomposition of organic matter (OM) exhibiting low decomposition rates should have a greater temperature sensitivity than that of faster-turnover substrates (see Laganière et al. 2015) while we observed no change in $Q_{10}$ with climate. Our results diverged from those of Laganière (2015) in that we did not find a lower reactivity to temperature, nor a greater amount of labile OM in the soils of the cold sites. The increase in labile organic matter on cold sites, observed in this latter study could be due to a greater abundance of bryophytes in the colder sites of this transect study, favouring the accumulation of labile organic matter (Kohl et al. 2018). We did not observe a trend for a greater abundance of bryophytes as a ground cover along our climatic gradient and this could explain the absence of a change in soil labile OM in our study. Bryophyte abundance is often linked to canopy openness (Pacé et al 2016) and our sites all had a closed canopy.

A reduction of labile soil C with warming had been observed in a manipulative experiment on a nearby site. D'orangeville et al. (2013) as well as Marty et al. (2019) found a reduction in active C with artificial soil warming, while they observed no change in soil organic matter composition congruently with our observations. The difference in the results from these studies and ours regarding changes in soil respiration reactivity to temperature ($Q_{10}$) and labile C, suggest that artificial soil warming may not have the full effect that long term climate warming has both on fluxes to and from the soil. A balance between C inputs and outputs may contribute to maintain soil organic matter stocks and properties despite warming. Climate gradient studies may also be more suited than warming experiments to observe changes in stable forms of soil organic matter because MAOM takes a long time to form. The fact that we did not observed differences in total SOM stocks in our study suggest that MAOM formation are not sensitive to climate perhaps because these reservoirs tend to saturate (Lavallée et al. 2020). Boreal regions show the highest concentrations of dissolved organic carbon (DOC) in the surface soil globally (Langeveld et al. 2020). This also suggest that the capacity for these soils to accumulate stable SOC is limited and that warming essentially changes the dynamics of unprotected POM.

Our results suggest that the dynamics of SOM that is not stabilised by association with minerals, that can be referred to as particulate organic matter (POM) has a large influence on C cycling in these soils. As indicated by Cotrufo et al. (2021), POM dominates the



dynamics of SOC cycling under cold and wet conditions. Apparently, warming both accelerated fluxes to and from the soil contribution to a stability of both SOM stocks and SOM stability.

## 5. CONCLUSION

Our results show no evidence of net SOM losses or a reduction of the most active SOM fraction with a warmer climate and contrast with conclusions reached in other studies derived from climate gradient studies or from direct soil warming experiments. Our results are applicable to the context of the study: a cold and wet environment where available moisture does not limit productivity or decomposition; closed canopy coniferous forests that were not recently disturbed and the absence of change in forest composition along the climate gradient. Other studies conducted under humid climate lend support to these results. Forest composition had effects on both SOM quality and rate of cycling. However, 
warming did not show interacting effects with forest composition. This suggest that the forest composition effect remains stable with changes in climatic conditions. Altogether, these results indicate that climate change effects on SOM storage and dynamics need to be studied both within and among forest ecosystem types.

## ACKNOWLEDGEMENTS

Financial support was provided by the PERD program of the Office of Energy Research and Development of Natural Resources Canada (Enhancement of Greenhouse Gas Sinks POL 6.2.1). We wish to thank L. St-Antoine, R. Morin, S. Audet and A. Courcelle for their contribution to field and lab work as well as Xiao Jing Guo, and Maryse Marchand for 
providing advice on the statistical analyses.

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

700

710

**Data Availability Statement**

The datasets generated for this study will be made available on Open Canada:

open.canada.ca/en