# Peer review of "Effects of a warmer climate and forest composition on soil carbon cycling, soil organic matter stability and stocks in a humid boreal region"

_EGUsphere, 2022_

## Author Comment (AC1)

Response to authors comments :

We thank the reviewer for their insightful and constructive comments! Comments and replies are copied here and also in the attached file which may be easier to read (color-coded).

Comments: The objective of the present study was to assess changes in SOC stocks, quality, and C fluxes to and from the soil along a climatic gradient occupied by two dominant and important stand types: balsam fir and black spruce. The more intensive aspects of the study leverage four black spruce and three balsam fir sites along the climate gradient, and flux measurements occurred between 2-4 years in duration, with some variation in frequency across the sites.  While the climate for the study region is deemed "humid" throughout, there appears to also be a gradient in precipitation (Table 1).  The data presentation and objectives are fairly straightforward and should be of broad interest to boreal ecologists.  There are a few areas that should be clarified in minor revision prior to publication, however. (I) There are some methodological aspects that were not clear to evaluate, at least to me.  It could be these are better explained in prior works by this team (e.g., is soil n only n=5, and is this enough power to assert changes?; why was a fixed value of 2 used for Q10, and yet Q10 was also determined directly?), but it would be good to clarify in this text. (II) There are now quite a few gradient studies examining C fluxes and soil C stocks in boreal conifer forests, including other work briefly mentioned in the text for Canada (for example, Boreal Forest Transect Case Study- Price and Apps), but also for Alaska and Fennoscandia, which would be great to discuss for context. This context would help in explaining co-variates with temperature along the gradient.  As such (III), it would be really nice if the authors could somehow evaluate the covariance of changes in precipitation and temperature along the climate gradient.  I believe these issues should be addressable in revision, and otherwise offer comments by line number, below, and hope they are helpful.

We thank the reviewer for their insightful and constructive comments!

1-Role of precipitation/aridity:

The design involved the selection of sites along a mean annual temperature gradient. We did not pay much attention to the role of precipitation because the study area is within a wet climatic region with few limitations of ecosystem processes due to water availability.

However, we concur with both reviewers that the role of precipitation should be considered more carefully because it is a global concern and we do have some potentially useful data to discuss this issue. This is what we propose: We will add an aridity index to the description of sites (Table 1). We used the Penman-Molteith equation as recommended by FAO (https://www.fao.org/3/x0490e/x0490e00.htm#Contents)  the ASCE standardized reference evapotranspiration (https://www.mesonet.org/images/site/ASCE_Evapotranspiration_Formula.pdf)

calculated daily from May to October over a 30 year period. We found out that for balsam fir sites only, there is a strong positive correlation between temperature (DD) and aridity: warmer sites are dryer (R2: 0.86; p<0.0001). This relationship is not significant for black spruce sites (R2=0.11; P=0.3537) as our cold black spruce sites included both wet (high elevation) and drier sites (high latitude). We explored the relationships between aridity and soil C stocks, litterfall, and soil respiration. Significant relationships were found only for fir sites between aridity and litterfall (a positive relationship: dryer = more productive). In addition, we also calculated RS10 (estimated soil respiration at 10oC i.e respiration adjusted for temperature) for each plot measurement event, generating about 4050 point measures. We compared RS10 with soil water content of the soil top 20cm assessed with a TDR probe. No relationship was found between soil moisture and respiration between sites or within the season. We also explored these relationships at the site level and found the same outcome. We will discuss this aspect in the paper and we will add the relationship in the form of graphs with statistical descriptions in the supplementary material.

In summary, for balsam fir sites, we cannot distinguish the impact of aridity from that of temperature, because both are strongly correlated. However, because we did not find any significant relationship between soil respiration and soil humidity measured in the field and because we do not find significant relationships between aridity and soil C stocks or soil C cycling for black spruce sites and for all sites together, we may conclude that aridity does not play a major role in controlling C stocks and C cycling under the wet climatic conditions of this study. We thank the author for this comment and we think that this addition will strengthen the paper. We will refer to Kane and Vogel (2009) and to Vogel et al. (2008) that are useful to better frame the context of our study. They found a reduction in soil C storage with warming past a certain threshold. However, these studies were conducted in a much drier climate. Our study region has an aridity index comparable to Amazonia (Trabucco, A., and Zomer, R.J. 2018. Global Aridity Index and Potential evapotranspiration (ET0) Climate Database v2). In addition, in Vogel et al. (2008) precipitation and temperature were positively correlated, while we observe the opposite. This gives support that the accelerated C fluxes and the absence of change in the soil C content that we observed with a warmer climate are likely the results of having no or little limitations of ecosystem processes by water. We will make this point clearer.

Line65: Appalachian Mountains?

2-the original text refers to the Southern Appalachians region

Table 1: The mean "annual precipitation" appears to differ by "site" along the gradients. Some statistical exploration of this would be good.

3-See comment # 1.

Line149: The "L" layer was not sampled?  I think this needs to be justified.  There could be big differences in the L layer (Oi soil horizon) in spruce vs. fir forests.

4-We only discarded the loose portion of the top litter because it is short-lasting and varies during the season. It represents a very small portion of the humus layer. We will add some precision to the text.

Line151: I don't understand: 5F+5H+5mineral is 15, and site n is still equal to 5.  Is n=5 sufficient to capture site-level variation for these systems, without bulking or taking composite samples?  For example, n=5 in Pare et al., 1993, but each "n" was the bulk product of 3 replicates (as such, 15 cores per site were taken). Ziegler et al. (2017) bulked 9 cores per site in their gradient study.

5-Recognizing that soil carbon stocks are highly variable at the plot level, we would like to stress that our study sites only covered an area of 400m2. Our sampling intensity is greater than what is used for national carbon inventories (NFI) and compared favorably well with scientific studies. We will make the sampling description clearer. (l. 150). We sampled 3 cores (not five as indicated) around each sample plot (5 per site). This generated 15 samples per soil layer (organic; 0-20cm and 20-40cm). Each sample was analyzed individually.

Line225: Excuse my ignorance, but I don't see why an assumed Q10 value would need to be used (contradictory to your equation 2, above)?

6-We estimated a Q10 value to compare sites along the climate gradient and between species. A Q10 of 2 was only used to interpolate the value of RS10 between measurement periods in the estimate of cumulative seasonal soil respiration (May to November). We will make this clearer in a new version. This methodology is derived from Lavigne et al. (2003). In fact, Lavigne et al. (2003) are citing four studies indicating that using a unique value of Q10 for the whole season can lead to overestimations. The rationale is that Q10 may change during the season. For example during periods of important root growth, it could be influenced by greater availability of root C to soil microbes. Nevertheless, the estimated respiration rate is the same on measurement day, regardless of the method used. It is only for the interpolation between measurement dates that they may slightly differ. In short, an RS10 is estimated for each site and measurement day with a Q10 of 2. This RS10 value (not Rs) is interpolated linearly between measurement days. To convert daily estimated RS10 to daily Rs values for no-data days, recorded soil temperature and a Q10 of 2 are used to back transfer Rs10 to Rs values.  Finally, and

recognizing that there is no standardized way of calculating these fluxes, we compared the two approaches, the one we used and the one using a Q10 that varies with site but that is the same for the whole season. The overall difference was 2% (the ratio of this second approach to the one we used ranged from 0.8 to 1.32, also suggesting a comparable but slightly skewed to higher values). We will refer to these results in the text and we would be happy to show the comparison in a table in the supplementary material.

As far as I can tell, it was Rayment and Jarvis (2000) who nominated the relatively consistent Q=2 for black spruce.  Since you are comparing across gradients and two dominant species cover types, I would recommend using your measured Q10, as in equation 2.

7-See above comment (6)

Line285: Regarding the assertion that there were no effects of "climate", does this include precipitation?  Can you be more specific?

 8-See above, we addressed this in point 1.

Table 3:  It would be so much better if precipitation was included in this analysis.

9-See above (1)

Figure 3: See for context, Kane ES, Valentine DW, Michaelson GJ, Fox JD, Ping C-L. 2006. Controls over pathways of carbon efflux from soils along climate and stand productivity gradients in interior Alaska. Soil Biology and Biochemistry. 38: 1438-1450.

Vogel et al. 2008. Carbon allocation in boreal black spruce forests across regions varying in soil temperature and precipitation. Global Change Biology.

 10-thanks for the suggestions, both similarities (increase NPP with temperature) and divergences  (declining soil stocks with temperature vs no change in our study) are found; it is interesting to note that the climate of these studies is dryer and this may explain the divergence. We will introduce them in the discussion.

Line371: Regarding the "uncertainty", I think it would be appropriate to have a nod to the low apparent power (soil n=5) for soil sampling in this study.

11-See above comment (5)

Line375: This may be true, but as stated, this is a bit of an over-simplification. As you state below, there are other factors varying here besides just "climate" in these studies. In Fennoscandia, the latitude gradient is confounded with N deposition. Moreover and as discussed in the Ziegler paper, precipitation co-varied with the climate gradient in Norris et al. 2011. Across climate gradients in AK which controlled for precipitation, and texture, and had similar N deposition, soil C declined with increasing soil growing degree days (Kane et al. 2005; Kane and Vogel, 2009). See also, earlier Vogel et al., 2008 GCB reference.

12-Interesting point! Two aspects, our climate is much wetter than those of that cited and in our study, at least for balsam fir, colder is wetter while in those studies colder was drier. (see point 10)

Line395: "We were not able to explain the large variability in soil C stocks across sites. This property is highly variable at a small scale and has notoriously been difficult to map (Paré et al. 2021). A much larger dataset would be required"

This is a very important point. If you are not capturing the variance at each site, can you assert that soil C stocks are truly not changing across the gradient? A quick power analysis could answer this question.

13-see (5) we think that our sampling intensity is adequate to capture within-site variability. However, we may add that what matters here is the large variability between sites.

Line405: "Both species also showed a stable litter C:N ratio along the climate gradient, suggesting

that the stoichiometry of C to N is not affected by climate.":

This is really interesting!

14-Thanks! However, we are not sure if we can make more of this observation; homeostasis in plant nutrition is common.

Line445: See earlier comment about Q10 being fixed at 2, and kindly disregard if I was off base there.

15-See comment above (6)

Line491: "Our results show no evidence of net SOM losses or a reduction of the most active SOM

fraction with a warmer climate" Can this be said, if the site level variance in SOM stocks is not being captured (vis a vis, line 395)?

16-see (5) and (13)

Response to authors comments :

We thank the reviewer for their insightful and constructive comments! Comments and replies are copied here and also in the attached file which may be easier to read (color-coded).

---

## Author Comment (AC2)

**General comments**

This study examines effects of climate and forest composition on soil organic carbon stocks in humid boreal forests of Canada. It uses elevation and latitude to create a climate gradient of forests spanning 4°C, dominated by balsam fir or black spruce trees. The authors found an effect of climate on carbon cycling (inputs and outputs) but no effect of climate on overall total soil organic carbon stocks (C stocks of the organic layer and top 40cm of the mineral soil), which is a result that is supported by some studies but not by others.

This study is important and the paper would be of interest to readers of Biogeosciences because of the large stocks of SOC that exist in boreal forests that we know are vulnerable to the rapid warming already occurring in northern ecosystems, however the mechanisms behind these C losses are not well understood and result in large uncertainties in modelling efforts. Furthermore, empirical measurements are needed to verify laboratory incubation results because the dominating controls determined in isolation in the laboratory are often difficult to observe in an intact system. This study is a strong contribution, therefore my criticisms are intended to strengthen the manuscript and provide "food for thought" for the authors.

The two larger scientific concerns I have are: 1) the metric used to evaluate the effects of climate is degree-days, and while there are instances where that is made explicitly clear it needs to be consistent throughout the manuscript. Climate is more than temperature, and climate change involves changes to precipitation as well as temperature. The authors nicely point out that the results of this study are applicable to "cold, humid" climates, however only the temperature component of climate change is tested, despite a 600+ mm range in precipitation across all the sites. If it is not possible to test MAP, I would like to see some info on soil moisture included at the very least;

We thank the reviewer for the insightful and constructive comments!

1-We have addressed this concern in length in the response to reviewer 1. Please see comment (1) in reply to reviewer 1.

and 2) Lability is a tricky concept that is measured in many different ways. This makes it difficult to compare between studies and interpret meaning. I challenge the use of mineralization as a measure of lability, especially in this study where lability is used as a potential explanation for Q10 variability (which is also respiration/temperature based). I don't necessarily think this part of the study should be removed but the caveats of the incubation as an indicator of lability should be discussed explicitly and critically. Also, Schmidt et al., 2011 suggests that even recalcitrant OM can be decomposed under the right environmental conditions, how do you know that labile OM is exclusively being mineralized in your incubations?

2-This is an interesting point! No, we do not know if some of the evolved C comes from recalcitrant forms and it may well be possible. We will change labile to bioavailable C as in Andrieux et al. (2020), the rationale being that we do not have information on the chemical nature of the organic material but rather on the potential for microbes to degrade it under standard conditions.

For the most part, this is a well prepared and presented manuscript. The figures and tables included are all useful, however some of them are blurry and difficult to read (Figures 3 and 4 in particular). There are several sentences in the text that require rewording, or reorganization. I've pointed out a few below. Some work is needed to make your hypotheses in the introduction clearer.

3-We will prepare figures with a better resolution and we will also move the concepts of organic matter reactivity in the text prior to stating the hypotheses.

**Specific comments**

Abstract

Line 12 "climate change is [a] matter of concern"

accepted

Line 19 "climate (cumulative degree days >5degreesC)", write like this throughout OR "climate (DD)" once DD is defined. Also, should mention somewhere in the manuscript why DD was chosen instead of MAT to represent climate

accepted

Line 25 change "spruce ones" to "spruce forests"

accepted

Line 28 "contrary to common soil organic matter stabilization hypotheses".  My intuitive thought is that greater cycling would result in increased losses and decreases in stocks, or is the assumption that labile portions get respired and the recalcitrant C is left behind and stabilised by minerals?

We will change the term cycling by inputs

Line 31 "apply to the context of this study: cold and wet environment". I appreciate that this statement was included, however not much has been done to address the "wet" part of that statement. Precipitation is variable (MAP: 954 - 1631 mm) and not tested, and no soil moisture data has been shown

We will add a metric of aridity and present results of the relationship between soil respiration and soil moisture, which was not significant. More details in response to Reviewer 1 (1)

Intro

Line 36, "Boreal forests should also experience the most intense warming" could be changed to "are experiencing the most intense warming"

accepted

Line 56, The sentence that starts with "However, because both C fluxes to and from the soil are accelerated by temperature..." has great points but the sentence took a while to process as written.

I suggest: "…the net effect of increased temperature on soil C accumulation will vary if the rates of input and output fluxes are differentially affected by temperature" or something like that

Accepted: much clearer!

Line 80 -84 This comment about wildfires, although important and relevant, is out of place here as your hypotheses have nothing to do with assessing the effects of wildfire on SOC stocks. Consider moving wildfire to the general climate change/ boreal section at the beginning of the intro if you want to keep it. This paragraph should have more info about litter quality differences between the two forest types and the effect on SOM, for instance.

Accepted

Hypothesis 1

Line 85: warmer sites accumulate more carbon? Is this reasonable given the greater driving hypothesis that climate warming = losses of SOC to the atmosphere? Can both be true? I think the mineral-associated OM and MEMS framework should is the part of the explanation that is missing and should be described in more detail before getting to the hypotheses here. Also isn't litter of higher quality (lower C:N, more labile) more easily decomposed and respired?

The framework developed by Cotrufo et al. (2013) was used as a cornerstone to define this hypothesis (more stable C) with greater C inputs. We elaborate in the discussion about the outcomes. We agree to move the description of the MEMS framework in a section that proceeds the hypotheses.

Line 90: can you clarify this point? I think I know what you mean, and I think it's related to my question above, but it needs to clearer. I like that the Andrieux, 2020 reference is included but I shouldn't need to go to that paper to understand the sentence. Is the point that the total (O.L. + mineral-associated to 40cm depth) carbon stock is important to capture? As opposed to studies that evaluate only O.L. stocks or only mineral C  stocks. Can the Andieux, 2020 paper be introduced in the main body of the intro before we get to the hypotheses? That might set things up better

Accepted: We will introduce the concept sooner in the text and this should make the hypotheses leaner. Andrieux et al. (2020) found that about 10% of the whole soil total organic C from the O horiozon to 40cm down the mineral soil could be qualified as fast C.

Hypothesis 3

Line 93: this is the hypothesis that I'm having trouble with. Is it fair to use mineralized losses (C and N mineralization) as the measure of labile carbon and nitrogen content, and then to use that data as an explanation for Q10 variability which is also respiration and temperature based? Shouldn't an independent measure of lability be considered? For instance, a chemical measure of lability? How do you know for sure that what is mineralized in the incubations is labile?

Materials and Methods

Line 115, do you have any quantitative measure of "closed-canopy"? This is brought up again in the discussion and I don't follow the logic with regard to bryophyte distribution.

We don't but open canopy boreal stands are easy to avoid and have an abundant understorey of bryophytes that may change the studied processes. Bryophytes, especially Sphagnum species and also lichen can change the soil microclimate and the decomposition process greatly (Pace et al. 2018 https://doi.org/10.1016/j.foreco.2018.02.020; Pace et al. 2020: https://doi.org/10.1007/s11104-020-04587-0)  . We wanted to avoid these situations.

Table 1, Please change annual precipitation to MAP

Accepted

Line 198, include simple description of the coefficients b1 and b2

Yes, we will: b1 is RS10 and b2 $=\ln(Q10)/10$ while $Q10=e^{10*b2}$; an error has slipped into the paper and Eq.3 is of no use and will be deleted.

Line 259, "depending on rates" why is this dependent on rates. Do rates reach zero? Please explain in the section.

We will add an explanation: we adapted the periods during which the lid was closed prior to $CO_2$ measurements to get concentrations that were within the range of calibration of the IRGA.

Line 262, was the nitrate and ammonium flushed to simulate field flushing of these species? Was this done monthly and why? Yes, this is done to maintain the soil humid and to flush the accumulation of metabolic products that may interfere with the decomposition process.

Line 265, how can you assume that what was mineralized was labile? Doesn't the Schmidt et al., 2011 reference suggest that even recalcitrant OM can be mineralized under the right environmental conditions? Couldn't recalcitrant OM be decomposed at 22C?

Incubation is an empirical method where we measure what the microbes are able to process under standard conditions. We will change labile to bioreactive. We will add in this section a clarification/definition of what we call labile C and N; which is not, as the reviewer rightly points out, a chemical definition.

Results

Line 279, instead of "this variability could not be attributed to a single factor" write, "this variability could not be attributed to species, DDS or their interaction (Table 2)"

accepted

Line 280, the sand comment seems out of place as soil texture is not mentioned anywhere else in the paper and was not tested.

Yes, we can remove the sentence. Coarse textured soils lead to little mineral-organic interactions and OM stabilization. But we did not explore these relationships and our design is poorly suited for this.

Line 282, use humus layer or organic layer but not both.

We will make sure that we have consistency in the terminology

Line 284, I appreciate that the OL and mineral C proportions are shown here, but no need to say that 33% is close to 25%. If the proportions are not significantly different between forest types then you should say that instead.

We agree!

Line 285, use DDs instead of climate in the results so that it is clear what is being used as a metric for climate.

Table 2: is Total C the sum of carbon in OL, 0-40cm, and coarse woody debris? This should be clear in the caption.

Line 302/309, stick with degree-days instead of climate, the two are used interchangeably in this paragraph and the next

Figure 2 is blurry

All of the above will be fixed!

Line 340, do you think differences in Q10 would be observed under a larger range in MAT (>4C)?

We don't know but we will make the data available for use in a larger gradient.

Line 344, replace "ones" with "soils"

Figure 4 is hard to read, blurry and small

This will be fixed

Discussion

Line 369, remove "in"

Accepted

Line 385, is there a relationship between MAT and MAP? Yes for fir only; See first comment to reviewer 1.

Line 386, it would be great to include the soil moisture data

Accepted

Line 387, "[Furthermore], the size of the SOM stock is not only controlled by climate or NPP, [but is also] strongly influenced by soil types…."

Accepted

Line 397, including MAP

Accepted

Line 403, add reference for needle statement

Added and figure adjusted

Line 419, is this because black spruce sites are already generally wetter than balsam fir?

Good point! I am not sure that they speculated on this but we can mention it as a supposition.

Line 437, replace "congruent results, that is to say" with "the"

Accepted

Line 466, this would be easier to interpret if there was more info on "closed-canopy"

We will add a sentence on the role of bryophytes on organic matter cycling.

Line 470, are you using "active" synonymously with labile? If so, just use labile for consistency

Yes, we will use a consistent terminology. We will use the term available as explained above. (available to microbes)

Line 475, "to maintain" should be "to the maintenance"

Line 477, this is first time we are seeing MAOM, please write it out in full

Line 483, this is the first time we are seeing POM, please write it out in full

Line 476 – 489, There are several points being made in this section with no clear connection. It is difficult to understand the connection between MAOM, DOC and POM and how it relates to your results. I would start this as a new paragraph and refine

We agree with the comments above; We will re-write, the last section as it is confusing. We will indicate that:  *Stabilized MAOM reservoirs may reach a saturation point, especially in non-recently disturbed soils (Lavallée et al. 2020). Boreal regions show the highest concentrations of dissolved organic carbon (DOC) in the surface soil globally (Langeveld et al. 2020), indicating that the capacity of these soils to immobilized DOC as water percolates through the soil column is limited.  Cotrufo et al. (2021) suggested that under cold and wet conditions, it is not MAOM but poorly stabilized particulate organic matter (POM) that dominates the dynamics of SOC cycling. If indeed MAOM reservoirs have reach saturation, and POM dominates the SOC cycling, our results suggest that warming, while accelerating SOC cycling does not lead to changes in the stocks of either POM or MAOM stocks. However, more research is needed to determine how the different fractions of SOM are impacted by changes both in aridity and in temperature and to identify climatic thresholds from which SOC stocks become vulnerable.*

Conclusion

Line 492, replace "active" with "labile" for consistency

See above, we will use bioreactive or reactive as in Andrieux et al. (2020)

Line 501, change "with changes in climate conditions" to "with projected changes to temperature" or something like that to tie it back to the climate change projections for the area

Accepted

Line 501- 503, I appreciate this final recommendation. Could expand it to include " these results indicate that climate change effects on SOM storage and dynamics need to be studied both within and among forest ecosystem types [in order to do what??]. How will continuing to do "within and among" studies help solve the problem? Please state explicitly. I think that would make for a more impactful ending!

Yes;  we will complete the sentence… in order to separate the direct effects of climate change from that of vegetation change.

---

## Author Response (AR1)

**Authors comments on a ms titled: Effects of a warmer climate and forest composition on soil carbon cycling, soil organic matter stability and stocks in a humid boreal region**

**July 19, 2022**

**Topical Editor's comments:**

Both of the international experts who reviewed your manuscript see merit in your study and feel your work can be an important contribution to the field. However, both reviewers also indicate several important points to be addressed. These includes some fundamental issues (e.g. what is the definition of labile soil C and what is an acceptable proxy to estimate this) that go beyond what could be addressed in minor revisions. Therefore, I will reconsider your manuscript upon major revisions along the lines of what you have already indicated in your extensive author response.

We would like to thank the Topical Editor and reviewers for their helpful comments! We have taken them all into account and believe the document has been greatly improved. Below is a point-by-point description of how we responded to the comments. Line number refer to the track change version.

In summary, the main changes are:

-Accounting for precipitation: we calculated aridity and examined the relationship between this factor and SOM stocks and the carbon cycle. Since the design was built to study a temperature gradient, inferences with aridity could not be fitted as well as those with temperature, and to avoid disrupting the flow of the text, most of the analyses were reported in the appendices. Aridity proved interesting as a potential explanation for SOC stocks and we think its inclusion is an interesting addition.

-Great care was taken in editing the text and standardizing the terminology and quality of the illustrations.

-Clarification on the use of the term "labile": In the previous version, we used the term labile to define organic matter that was not stabilised and therefore prone to mineralisation. We agree that this term can be misleading as it often refers to the chemical property of SOM. According to He and Wu (2015) (He and Wu 2015: Labile organic matter, SSSA Special publication 62) the term "labile" *usually refers to SOM that has a fast turnover and that includes, but is not limited to light fractions, easily extractable pools, microbial biomass C and labile humic materials*. We prefer to use the term bioreactive SOM as we did in an earlier publication in SOIL (Andrieux et al. doi.org/10.5194/soil-6-195-2020) and we made this change throughout the document. We think that this terminology is more appropriate. It refers to SOM that is prone to mineralization under standard laboratory conditions and we do not make any inferences to the age or the chemical nature of this material. We have carefully evaluated the use of this term everywhere in the text.

**Response to authors comments:**

We thank the reviewer for their insightful and constructive comments! Comments and replies are copied here.

Reviewer 1:

Comments: The objective of the present study was to assess changes in SOC stocks, quality, and C fluxes to and from the soil along a climatic gradient occupied by two dominant and important stand types: balsam fir and black spruce. The more intensive aspects of the study leverage four black spruce and

three balsam fir sites along the climate gradient, and flux measurements occurred between 2-4 years in duration, with some variation in frequency across the sites. While the climate for the study region is deemed "humid" throughout, there appears to also be a gradient in precipitation (Table 1). The data presentation and objectives are fairly straightforward and should be of broad interest to boreal ecologists. There are a few areas that should be clarified in minor revision prior to publication, however. (I) There are some methodological aspects that were not clear to evaluate, at least to me. It could be these are better explained in prior works by this team (e.g., is soil n only n=5, and is this enough power to assert changes?; why was a fixed value of 2 used for Q10, and yet Q10 was also determined directly?), but it would be good to clarify in this text. (II) There are now quite a few gradient studies examining C fluxes and soil C stocks in boreal conifer forests, including other work briefly mentioned in the text for Canada (for example, Boreal Forest Transect Case Study- Price and Apps), but also for Alaska and Fennoscandia, which would be great to discuss for context. This context would help in explaining co-variates with temperature along the gradient. As such (III), it would be really nice if the authors could somehow evaluate the covariance of changes in precipitation and temperature along the climate gradient. I believe these issues should be addressable in revision, and otherwise offer comments by line number, below, and hope they are helpful.

1-Role of precipitation/aridity:

The experimental design was based on the selection of sites along a mean annual temperature gradient. We overlooked the role of precipitation because the study area is within a wet climatic region with few limitations of ecosystem processes due to water availability. However, we concur with both reviewers that the role of precipitation should be considered more carefully because it is a global concern and we do have some potentially useful data to discuss this issue. We have added an aridity index to the description of sites (Table 1). We used the Penman-Molteith equation as recommended by FAO (https://www.fao.org/3/x0490e/x0490e00.htm#Contents) the ASCE standardized reference evapotranspiration (https://www.mesonet.org/images/site/ASCE_Evapotranspiration_Formula.pdf) calculated daily from May to October over a 30-year period. We found out that for balsam fir sites only, there is a strong positive correlation between temperature (DD) and aridity: warmer sites are dryer ($R^2$: 0.86; $p<0.0001$). This relationship is not significant for black spruce sites ($R^2=0.11$; $P=0.3537$) as our cold black spruce sites included both wet (high elevation) and drier sites (high latitude). Details are provided in SM1. We then explored the relationships between aridity and soil C stocks and fluxes:

-A significant negative relationship was found between mineral soil SOM stocks and aridity for spruce stands but not for balsam fir stand. This is an interesting addition to the paper as it suggests that aridity, not temperature (DD) is limiting SOC stocks. We added information about this relationship in the supplementary material (SM6) and we discuss this aspect in the discussion (l.570).

-Significant relationships were also found between aridity and C fluxes: the litterfall rate of balsam fir sites as well as total soil respiration for all sites (RS) were positively related to aridity (dryer = more productive). These relationships suggest that water is abundant enough on the warmest sites that it does prevent the relationship between DD and C cycling rates to take place. Text was added (l. 640).

We now refer to Kane and Vogel (2009) that is useful to better frame the context of our study. They found a reduction in soil C storage with warming. However, these studies were conducted in a much drier climate. Our study region has an aridity index comparable to Amazonia or to the Pacific Northwest (Trabucco, A., and Zomer, R.J. 2018. Global Aridity Index and Potential evapotranspiration (ET0) Climate Database v2). In addition, in Vogel et al. (2008) precipitation and temperature were positively correlated, while we observed the opposite. This suggests that a negative effect of climate warming on SOM stocks appears above a certain threshold of aridity (l.564-70).

Line65: Appalachian Mountains?

2-the original text refers to the Southern Appalachians region (change made L. 78)

Table 1: The mean "annual precipitation" appears to differ by "site" along the gradients. Some statistical exploration of this would be good.

3-See comment # 1.

Line149: The "L" layer was not sampled? I think this needs to be justified. There could be big differences in the L layer (Oi soil horizon) in spruce vs. fir forests.

4-We only discarded the loose portion of the top litter because it is short-lasting and varies during the season. It represents a very small portion of the humus layer. Details were added (l.263).

Line151: I don't understand: 5F+5H+5mineral is 15, and site n is still equal to 5. Is n=5 sufficient to capture site-level variation for these systems, without bulking or taking composite samples? For example, n=5 in Pare et al., 1993, but each "n" was the bulk product of 3 replicates (as such, 15 cores per site were taken). Ziegler et al. (2017) bulked 9 cores per site in their gradient study.

5 Recognizing that soil carbon stocks are highly variable at the plot level, we would like to stress that our study sites only covered an area of $400m^2$. Our sampling intensity is greater than what is used for national carbon inventories (NFI) and compared favorably well with scientific studies. We have made the sampling description clearer. (l. 234-235). We sampled 3 cores (not five as previously indicated) around each sample plot (5 per site). This generated 15 samples per soil layer (organic; 0-20cm and 20-40cm). Each sample was analyzed individually.

Line225: Excuse my ignorance, but I don't see why an assumed Q10 value would need to be used (contradictory to your equation 2, above)?

6-We estimated a Q10 value to compare sites along the climate gradient and between species. A Q10 of 2 was only used to interpolate the value of RS10 between measurement periods in the estimate of cumulative seasonal soil respiration (May to November). We will make this clearer in a new version. This methodology is derived from Lavigne et al. (2003). In fact, Lavigne et al. (2003) are citing four studies indicating that using a unique value of Q10 for the whole season can lead to overestimations. The rationale is that Q10 may change during the season. For example during periods of important root growth, it could be influenced by greater availability of root C to soil microbes. Nevertheless, the estimated respiration rate is the same on measurement day, regardless of the method used. It is only for the interpolation between measurement dates that they may slightly differ. In short, an RS10 is estimated for each site and measurement day with a Q10 of 2. This RS10 value (not Rs) is interpolated linearly between measurement days. To convert daily estimated RS10 to daily Rs values for no-data days, recorded soil temperature and a Q10 of 2 are used to back transfer Rs10 to Rs values. Finally, and recognizing that there is no standardized way of calculating these fluxes, we compared the two approaches, the one we used and the one using a Q10 that varies with site but that is the same for the whole season. The overall difference was of 1%). We refer to these results in the text and we present the results of this comparison in SM2. (l.360)

As far as I can tell, it was Rayment and Jarvis (2000) who nominated the relatively consistent Q=2 for black spruce. Since you are comparing across gradients and two dominant species cover types, I would recommnd using your measured Q10, as in equation 2.

7-See above comment (6)

Line285: Regarding the assertion that there were no effects of "climate", does this include precipitation? Can you be more specific?

8-See above, we addressed this in point 1.

Table 3: It would be so much better if precipitation was included in this analysis.

9-See above (1)

Figure 3: See for context, Kane ES, Valentine DW, Michaelson GJ, Fox JD, Ping C-L. 2006. Controls over pathways of carbon efflux from soils along climate and stand productivity gradients in interior Alaska. Soil Biology and Biochemistry. 38: 1438-1450.

Vogel et al. 2008. Carbon allocation in boreal black spruce forests across regions varying in soil temperature and precipitation. Global Change Biology.

10-thanks for the suggestions, both similarities (increase NPP with temperature) and divergences (declining soil stocks with temperature vs no change in our study) are found; it is interesting to note that the climate of these studies is dryer and this may explain the divergence. See comment 1.

Line371: Regarding the "uncertainty", I think it would be appropriate to have a nod to the low apparent power (soil n=5) for soil sampling in this study.

11-See above comment (5)

Line375: This may be true, but as stated, this is a bit of an over-simplification. As you state below, there are other factors varying here besides just "climate" in these studies. In Fennoscandia, the latitude gradient is confounded with N deposition. Moreover and as discussed in the Ziegler paper, precipitation co-varied with the climate gradient in Norris et al. 2011. Across climate gradients in AK which controlled for precipitation, and texture, and had similar N deposition, soil C declined with increasing soil growing degree days (Kane et al. 2005; Kane and Vogel, 2009). See also, earlier Vogel et al., 2008 GCB reference.

12-Interesting point! Two aspects, our climate is much wetter than those of that cited and in our study, and at least for balsam fir, colder is wetter while in those studies colder was drier. (see point 10)

Line395: "We were not able to explain the large variability in soil C stocks across sites. This property is highly variable at a small scale and has notoriously been difficult to map (Paré et al. 2021). A much larger dataset would be required"

This is a very important point. If you are not capturing the variance at each site, can you assert that soil C stocks are truly not changing across the gradient? A quick power analysis could answer this question.

13-see (5) we think that our sampling intensity is adequate to capture within-site variability. However, we may add that what matters here is the large variability between sites.

Line405: "Both species also showed a stable litter C:N ratio along the climate gradient, suggesting

that the stoichiometry of C to N is not affected by climate.":This is really interesting!

14-Thanks! However, we are not sure if we can make more of this observation; homeostasis in plant nutrition is common.

Line445: See earlier comment about Q10 being fixed at 2, and kindly disregard if I was off base there.

15-See comment above (6)

Line491: "Our results show no evidence of net SOM losses or a reduction of the most active SOM

fraction with a warmer climate" Can this be said, if the site level variance in SOM stocks is not being captured (vis a vis, line 395)?

16-see (5) and (13)

**Reviewer 2**

**General comments**

This study examines effects of climate and forest composition on soil organic carbon stocks in humid boreal forests of Canada. It uses elevation and latitude to create a climate gradient of forests spanning 4°C, dominated by balsam fir or black spruce trees.  The authors found an effect of climate on carbon cycling (inputs and outputs) but no effect of climate on overall total soil organic carbon stocks (C stocks of the organic layer and top 40cm of the mineral soil), which is a result that is supported by some studies but not by others.

This study is important and the paper would be of interest to readers of Biogeosciences because of the large stocks of SOC that exist in boreal forests that we know are vulnerable to the rapid warming already occurring in northern ecosystems, however the mechanisms behind these C losses are not well understood and result in large uncertainties in modelling efforts. Furthermore, empirical measurements are needed to verify laboratory incubation results because the dominating controls determined in isolation in the laboratory are often difficult to observe in an intact system. This study is a strong contribution, therefore my criticisms are intended to strengthen the manuscript and provide "food for thought" for the authors.

The two larger scientific concerns I have are: 1) the metric used to evaluate the effects of climate is degree-days, and while there are instances where that is made explicitly clear it needs to be consistent throughout the manuscript. Climate is more than temperature, and climate change involves changes to precipitation as well as temperature.  The authors nicely point out that the results of this study are applicable to "cold, humid" climates, however only the temperature component of climate change is tested, despite a 600+ mm range in precipitation across all the sites. If it is not possible to test MAP, I would like to see some info on soil moisture included at the very least;

We thank the reviewer for the insightful and constructive comments!

1-We have addressed this concern in length in the response to reviewer 1. Please see comment (1) in reply to reviewer 1.

 and 2) Lability is a tricky concept that is measured in many different ways. This makes it difficult to compare between studies and interpret meaning. I challenge the use of mineralization as a measure of lability, especially in this study where lability is used as a potential explanation for Q10 variability (which is also respiration/temperature based). I don't necessarily think this part of the study should be removed but the caveats of the incubation as an indicator of lability should be discussed explicitly and critically. Also, Schmidt et al., 2011 suggests that even recalcitrant OM can be decomposed under the right environmental conditions, how do you know that labile OM is exclusively being mineralized in your incubations?

2-This is an interesting point! No, we do not know if some of the evolved C comes from recalcitrant forms and it may well be possible. We have change labile to bioavailable C as in Andrieux et al. (2020), the rationale being that we do not have information on the chemical nature of the organic material but rather on the potential for microbes to degrade it under standard conditions. See also response to the Topical Editor.

For the most part, this is a well prepared and presented manuscript. The figures and tables included are all useful, however some of them are blurry and difficult to read (Figures 3 and 4 in particular). There are several sentences in the text that require rewording, or reorganization. I've pointed out a few below. Some work is needed to make your hypotheses in the introduction clearer.

3-We have prepared figures with a better resolution and we will also move the concepts of organic matter reactivity in the text prior to stating the hypotheses (l.83-90)

**Specific comments**

Abstract

Line 12 "climate change is [a] matter of concern"

4-The change has been made.

Line 19 "climate (cumulative degree days >5degreesC)", write like this throughout OR "climate (DD)" once DD is defined. Also, should mention somewhere in the manuscript why DD was chosen instead of MAT to represent climate

5-The change has been made.

Line 25 change "spruce ones" to "spruce forests"

6-The change has been made.

Line 28 "contrary to common soil organic matter stabilization hypotheses".  My intuitive thought is that greater cycling would result in increased losses and decreases in stocks, or is the assumption that labile portions get respired and the recalcitrant C is left behind and stabilised by minerals?

7-We will change the term cycling by inputs. The text is now consistent with the the Microbial Efficiency-Matrix Stabilization (MEMS) framework developed in Cotrufo et al. (2013).

Line 31 "apply to the context of this study: cold and wet environment". I appreciate that this statement was included, however not much has been done to address the "wet" part of that statement. Precipitation is variable (MAP: 954 - 1631 mm) and not tested, and no soil moisture data has been shown

8-We will add a metric of aridity and present results of the relationship between soil respiration and soil moisture, which was not significant. More details in response to Reviewer 1 (1)

Intro

Line 36, "Boreal forests should also experience the most intense warming" could be changed to "are experiencing the most intense warming"

9-Accepted, the change has been made.

Line 56, The sentence that starts with "However, because both C fluxes to and from the soil are accelerated by temperature..." has great points but the sentence took a while to process as written.

I suggest: "...the net effect of increased temperature on soil C accumulation will vary if the rates of input and output fluxes are differentially affected by temperature" or something like that

10-Accepted: much clearer! The change has been made.

Line 80 -84 This comment about wildfires, although important and relevant, is out of place here as your hypotheses have nothing to do with assessing the effects of wildfire on SOC stocks. Consider moving wildfire to the general climate change/ boreal section at the beginning of the intro if you want to keep it. This paragraph should have more info about litter quality differences between the two forest types and the effect on SOM, for instance.

11-We wish to keep information on the ecology of the two species in the introduction. Information about the difference in litter quality and its implication is in the results and discussion sections. This sentence is to indicate that in our study area, forest composition is linked with the disturbance regime and that a warmer climate would likely reduce the area of balsam fir as it is very slow to re-colonise following fire. We simply wanted to give the context of why we think we should pay attention to forest tree composition in a climate change context. We don't intent to make any inferences to the direct role of fire on C cycling.

Hypothesis 1

Line 85: warmer sites accumulate more carbon? Is this reasonable given the greater driving hypothesis that climate warming = losses of SOC to the atmosphere? Can both be true? I think the mineral-associated OM and MEMS framework should is the part of the explanation that is missing and should be described in more detail before getting to the hypotheses here. Also isn't litter of higher quality (lower C:N, more labile) more easily decomposed and respired?

12-The framework developed by Cotrufo et al. (2013) was used as a cornerstone to define this hypothesis (more stable C) with greater C inputs. We elaborate in the discussion about the outcomes. We agree to move the description of the MEMS framework in a section that proceeds the hypotheses (l.83-90).

Line 90: can you clarify this point? I think I know what you mean, and I think it's related to my question above, but it needs to clearer. I like that the Andrieux, 2020 reference is included but I shouldn't need to go to that paper to understand the sentence. Is the point that the total (O.L. + mineral-associated to 40cm depth) carbon stock is important to capture? As opposed to studies that evaluate only O.L. stocks

or only mineral C  stocks. Can the Andieux, 2020 paper be introduced in the main body of the intro before we get to the hypotheses? That might set things up better

13-Accepted: We have introduced the concept sooner in the text and this should make the hypotheses clearer. Andrieux et al. (2020) found that about 10% of the whole soil total organic C from the O horizon down to 40cm in the mineral soil could be qualified as fast C.

Hypothesis 3

Line 93: this is the hypothesis that I'm having trouble with. Is it fair to use mineralized losses (C and N mineralization) as the measure of labile carbon and nitrogen content, and then to use that data as an explanation for Q10 variability which is also respiration and temperature based? Shouldn't an independent measure of lability be considered? For instance, a chemical measure of lability? How do you know for sure that what is mineralized in the incubations is labile?

14-We now use the term bioreactive instead of labile. More details in reply to the Topical Editor.

Materials and Methods

Line 115, do you have any quantitative measure of "closed-canopy"? This is brought up again in the discussion and I don't follow the logic with regard to bryophyte distribution.

15-We do not have this information. However, open canopy boreal stands are easy to avoid and have an abundant understorey of bryophytes that may change the studied processes. Bryophytes, especially Sphagnum species and also lichen can change the soil microclimate and the decomposition process greatly. We wanted to avoid these situations. We added a citation and a better context in the discussion l. 699.

Table 1, Please change annual precipitation to MAP

16-The change has been made.

Line 198, include simple description of the coefficients b1 and b2

17-Yes, we will: b1 is RS10 and b2 =ln(Q10)/10 while Q10=$e^{10*b2}$; We noticed that an error has slipped into the paper and Eq.3 is of no use and will be deleted.

Line 259, "depending on rates" why is this dependent on rates. Do rates reach zero? Please explain in the section.

18-We rephrased this section: *Soil respiration rates were measured monthly during periods of 4 to 24h during which the lid was closed. The length of these periods depended on rates and CO2 evolution in order to keep concentrations within the calibration range of the IRGA, a LI-6400 portable photosynthesis system (LI-COR, Lincoln, NE, USA). L. 395-400.*

Line 262, was the nitrate and ammonium flushed to simulate field flushing of these species? Was this done monthly and why?

19. Flushing is done to maintain the soil humid and to flush the accumulation of metabolic products that may interfere with the decomposition process. (l401-403)

Line 265, how can you assume that what was mineralized was labile? Doesn't the Schmidt et al., 2011 reference suggest that even recalcitrant OM can be mineralized under the right environmental conditions? Couldn't recalcitrant OM be decomposed at 22C?

20. Incubation is an empirical method where we measure the net products of microbial activities under standard conditions. We will change labile to bioreactive which is not, as the reviewer rightly points out, a chemical definition.

Results

Line 279, instead of "this variability could not be attributed to a single factor" write, "this variability could not be attributed to species, DDS or their interaction (Table 2)"

21-The change has been made.

Line 280, the sand comment seems out of place as soil texture is not mentioned anywhere else in the paper and was not tested.

22-We have removed the sentence. Coarse textured soils lead to little mineral-organic interactions and OM stabilization. However, we did not explore these relationships and our design is poorly suited for this.

Line 282, use humus layer or organic layer but not both.

23-We now use organic layer (4 changes)

Line 284, I appreciate that the OL and mineral C proportions are shown here, but no need to say that 33% is close to 25%. If the proportions are not significantly different between forest types then you should say that instead.

24- We did not test statistically for the proportion of C in the different layers. We are hesitant at testing for a large number of factors on a relatively small dataset.  We would like to keep the text as it is because it gives general indications that the proportion of OC contained in the organic layer is not that different between the two forest types. We did not include wetland black spruce sites which are very common in the boreal and which would have increase the ratio. This sentence is simply to inform that the two forest types should have relatively similar site conditions.

Line 285, use DDs instead of climate in the results so that it is clear what is being used as a metric for climate.

25-We agree that DD is more precise than climate and we changed the wording although the document.

Table 2: is Total C the sum of carbon in OL,  0-40cm, and coarse woody debris? This should be clear in the caption.

25-changes made

Line 302/309, stick with degree-days instead of climate, the two are used interchangeably in this paragraph and the next

26-done

Figure 2 is blurry

27-we will make sure that the quality meets the publication requirement in the final submission.

Line 340, do you think differences in Q10 would be observed under a larger range in MAT (>4C)?

28-We can't speculate on this but we will make the data available for other users to combine with other data and generate a larger gradient.

Line 344, replace "ones" with "soils"

29-The change has been made

Figure 4 is hard to read, blurry and small

30-we fixed this!

Discussion

Line 369, remove "in"

31-The change has been made.

Line 385, is there a relationship between MAT and MAP?

32-Yes for fir only; See first comment to reviewer 1.

Line 386, it would be great to include the soil moisture data

33-We have now include data on aridity (Table 1) and have discuss the implications throughout the text.

Line 387, "[Furthermore], the size of the SOM stock is not only controlled by climate or NPP, [but is also] strongly influenced by soil types…."

34-The change has been made

Line 397, including MAP

35-We included the term aridity

Line 403, add reference for needle statement

36-Added and the numbers were adjusted. (l.612-14)

Line 419, is this because black spruce sites are already generally wetter than balsam fir?

37-We think that this is related to the greater evapotranspiration needs of balsam fir but we have no data to support this.

Line 437, replace "congruent results, that is to say" with "the"

38- Accepted

Line 466, this would be easier to interpret if there was more info on "closed-canopy"

-39 We added a sentence and a citation on the role of bryophytes on organic matter cycling. (l. 698)

Line 470, are you using "active" synonymously with labile? If so, just use labile for consistency

-40 we use the term bioreactive for our experiment and we used the terminology that was used in the studies that were cited, in this case labile. (l.715-740)

Line 475, "to maintain" should be "to the maintenance"

41- the change was made.

Line 477, this is first time we are seeing MAOM, please write it out in full

42- the change was made.

Line 483, this is the first time we are seeing POM, please write it out in full

43- the change was made.

Line 476 – 489, There are several points being made in this section with no clear connection. It is difficult to understand the connection between MAOM, DOC and POM and how it relates to your results. I would start this as a new paragraph and refine

44-We agree with this comment! The section was re-written. (l. 785-795)

Conclusion

Line 492, replace "active" with "labile" for consistency

45-ee above, we will use bioreactive or reactive as in Andrieux et al. (2020)

Line 501, change "with changes in climate conditions" to "with projected changes to temperature" or something like that to tie it back to the climate change projections for the area

46-the change was made

Line 501- 503, I appreciate this final recommendation. Could expand it to include " these results indicate that climate change effects on SOM storage and dynamics need to be studied both within and among forest ecosystem types [in order to do what??]. How will continuing to do "within and among" studies help solve the problem? Please state explicitly. I think that would make for a more impactful ending!

47-we have rewritten this sentence.